# Correlational Study of Emotional Stress, Pain, and the Presence of Inpatient Companions for Cancer Inpatients during the COVID-19 Pandemic

**DOI:** 10.3390/ijerph19127004

**Published:** 2022-06-08

**Authors:** Ya-Huei Chen, Shu-Ling Chen, Chia-Hui Chang, Pi-O Wu, Hsiu-Hui Yu, Sou-Jen Shih, Mei-Yu Chang

**Affiliations:** 1Department of Nursing, Taichung Veterans General Hospital, Taichung 40705, Taiwan; chyah@vghtc.gov.tw (Y.-H.C.); jeans@vghtc.gov.tw (C.-H.C.); nx8288@vghtc.gov.tw (P.-O.W.); f7741@vghtc.gov.tw (H.-H.Y.); soujen@vghtc.gov.tw (S.-J.S.); cmy6040@vghtc.gov.tw (M.-Y.C.); 2Department of Nursing, Hungkuang University, Taichung 433304, Taiwan

**Keywords:** coronavirus disease 2019 (COVID-19), pandemic, cancer inpatient, inpatient companion

## Abstract

The outbreak of COVID-19 poses an immense global threat. Visitors to hospitalized patients during a pandemic might themselves be carriers, and so hospitals strictly control patients and inpatient companions. However, it is not easy for cancer patients to adjust the times of their medical treatment or to suspend treatment, and the impact of the pandemic on cancer inpatients and inpatient companions is relatively high. The objectives for this investigation are to study the correlations among emotional stress, pain, and the presence of inpatient companions in cancer patients during the COVID-19 pandemic. This study was a retrospective descriptive study. The participants were cancer inpatients and inpatient companions in a medical center in Taiwan. The data for this study were extracted from cross-platform structured and normalized electronic medical record databases. Microsoft Excel 2016 and SPSS version 22.0 were used for analysis of the data. In all, 75.15% of the cancer inpatients were accompanied by family, and the number of hospitalization days were 7.87 ± 10.77 days, decreasing year by year, with statistical significance of *p* < 0.001. The daily nursing hours were 12.94 ± 10.76, and the nursing hours decreased year by year, *p* < 0.001. There was no significant difference in gender among those who accompanied the patients, but there were statistical differences in the length of hospitalization, nursing hours, and pain scores between those with and without inpatient companions, with *p* < 0.001. The inpatient companions were mostly family members (78%). The findings of this study on cancer patient care and inpatient companions should serve as an important basis for the transformation and reform of the inpatient companion culture and for epidemic prevention care in hospitals.

## 1. Introduction

In 2020, COVID-19 became a global pandemic [1]. The number of cases in Taiwan remained relatively stable compared with those in other countries. However, in February, 2021, the number of domestic cases began to rise. From May to August, the whole country was under local level 3 restrictions [2], which closed down schools and offices. Patients with cancer are a vulnerable population, and during such pandemics, they may experience life-threatening infections and interruption of their cancer or usual medical care [3]. The timing of the medical treatment of cancer inpatients cannot easily be adjusted or delayed, and the impact of an epidemic on cancer inpatients and their families is relatively high [4,5]. However, Taiwan’s long-standing “inpatient companion culture” is different from that in advanced countries, where hospital professionals are solely responsible for patient care and only allow regular visits to patients [6,7]. Most hospitals in Taiwan inform inpatients that they need to be accompanied by inpatient companions so that during treatments or emergencies, the hospital can communicate with the patient’s family in a timely manner. The inpatient companion can also accompany the patient and share the care work so as to reduce the workload of clinical care [8,9]. However, during the COVID-19 pandemic, national policy stipulated that only one inpatient companion could accompany the patient. In addition, the hospital’s supporting measures for the management of inpatient companions advocated the reduction of frequent rotations of inpatient companions, and when no inpatient companion was available, the nurses would fully intervene in the care work so as to reduce the risk of improper management or inpatient companions causing nosocomial infections [9]. However, such restrictions could have serious impacts on the physical and mental conditions of cancer inpatients. According to the American Cancer Society, primary inpatient companions provide an average of 8.3 h of care per day [10], while domestic primary inpatient companions provide an average of 16–24 h of care per day [4]. The number of hours of care provided by the inpatient companion may affect the physical and mental condition of the patient. Relevant studies in the United States have pointed out that the primary inpatient companion provides more than 10 h of nursing activities per week on average, known as the care load, and more than 35 h per week (average 5 h per day) is considered a high load [11]. If the number of visitors is limited to one person and frequent inpatient companion rotations are not recommended, the care load of the inpatient companion is higher [12]. During the COVID-19 pandemic in Taiwan, those who entered hospitals to visit patients were required to register. As of 28 May 2021, the first inpatient companion to accompany a patient in the hospital was required to have a negative result on a publicly funded polymerase chain reaction (PCR) test. The second inpatient companion had to pay NTD 2500 (about USD 85) out of pocket for the PCR test [13,14]. Family members usually care for a patient and satisfy the patient’s needs for love and a sense of belonging. However, during the pandemic, family visits were forbidden, and only one inpatient companion could be present, which greatly changed the culture of hospital inpatient companions in Taiwan.

For the hospital, there were some benefits. They could implement a registration system for inpatient companions, control the number of people entering and leaving the wards, and reduce the likelihood of cluster breeding; in addition, the doctor could communicate with only one inpatient companion. However, being in close contact with a patient may aggravate the fatigue of the inpatient companion. For example, inpatient companions have no time to deal with personal problems and no privacy, and they also need to follow the hospital’s pandemic prevention regulations. In addition, they also face the fear of infection and stress of being questioned by family members about changes in the patient’s condition [6]. Therefore, the purpose of this study was to investigate and analyze the profiles of cancer inpatients and their companions during the COVID-19 pandemic and to explore (1) the demographic characteristics, pain scores, and emotional stress levels of cancer inpatients during the COVID-19 pandemic; (2) the experience of the inpatient companions; and (3) the correlations among the demographic characteristics, pain scores, and emotional stress levels of cancer inpatients and their inpatient companions.

## 2. Method

### 2.1. Research Design

We retrospectively analyzed data. All the study procedures were approved by the Hospital Human Investigation Committee (IRB No. CE21331A), and the included participants were cancer inpatients over 20 years old and their inpatient companions. Data from 1–30 June of the years 2019, 2020, and 2021 were collected from the Clinical Data Center, a survey of the medical record database, and the inpatient companion registration system. A total of 3103 patients and 2332 inpatient companions from before and during the COVID-19 pandemic were investigated, and the relevant data were statistically analyzed.

### 2.2. Measurements

#### 2.2.1. The Numeric Rating Scale

The Numeric Rating Scale (NRS) is an 11-point scale of 0 to 10, with 0 indicating “no pain” and 10 indicating the “worst imaginable pain”. Patients are instructed to choose a single number on the scale that best indicates their level of pain. The NRS is simple and has been found to be a valid measure of pain intensity [15,16].

#### 2.2.2. Brief Symptom Rating Scale (BSRS-5)

The BSRS-5 is a five-item self-report questionnaire in which a higher score indicates poorer mental health. The full scale contains the following five items related to psychopathology: (1) feeling tense or keyed up (anxiety); (2) feeling blue (depression); (3) feeling easily annoyed or irritated (hostility); (4) feeling inferior to others (inferiority); and (5) having trouble falling asleep (insomnia). An additional question, “Do you have any suicidal ideation?” was added at the end of the questionnaire. The subjects were asked to rate symptoms on a five-point Likert-type scale as follows: 0, not at all; 1, a little bit; 2, moderately; 3, quite a bit; and 4, extremely. A total score was calculated for each subject. The BSRS-5 has demonstrated good reliability and validity [17,18]. The internal consistency (Cronbach’s alpha) coefficient of the BSRS-5 was 0.84 [19]. The BSRS-5, either self-rated or administered by interview, has been reported to have satisfactory psychometric properties for detecting psychiatric morbidity in medical practice and the community. It has been used widely in various settings as a screening tool in Taiwan, where it is nick-named the “mood thermometer” [17,18,19].

### 2.3. Statistical Analysis

Microsoft Excel 2016 and SPSS version 22.0 (SPSS Inc., Chicago, IL, USA) were used for data analysis and verification of the non-normal distribution of the data. Continuous data are non-normally distributed, and results are presented as means and standard deviations. The inferential statistics are based on the Kruskal–Wallis test and chi-squared test to test the difference between the basic data of the inpatients and their inpatient companions from 1–30 June in 2019, 2020, and 2021, and the post hoc testing used the Dunn–Bonferroni method. For the inpatients and their inpatient companions, the Mann–Whitney U test and chi-squared test were used.

## 3. Results

### 3.1. Demographic Data of Cancer Inpatients and Inpatient Companions

In the three periods, there were 3103 cancer inpatients and 2332 inpatient companions (Table 1). From 1–30 June of the years 2019, 2020, and 2021, the total numbers of daily cancer inpatients at the study hospitals were 975, 1179, and 949, respectively. There were 1459 males (47.02%) and 1644 females (52.98%). The average age of the patients was 59.71 ± 12.66 years, and there was a statistical difference in the three periods; *p* < 0.001. The average length of hospital stay was 7.87 ± 10.77 days, with a statistical difference in the three periods of *p* < 0.001. The average length of stay in the three years decreased year by year, from 8.29 days to 7.69 days and 7.67 days, respectively. Our study conducted further analysis through post hoc tests. First, for age, it can be found that the average age of received cases in 2021 was younger than those in 2019 and 2020, with statistical significance (*p* < 0.001). Second, for the number of days in hospital, the numbers of hospitalization days in different years were different. The total in 2019 was much higher than those in 2020 and 2021. The numbers of inpatients who had companions in the three periods were 605, 987, and 740, respectively. The pain scores were ≥4, and there was no statistical difference in the three periods. (Table 1). Nursing hours were 12.94 ± 10.76; *p* < 0.001. The total nursing hours in the three periods increased by 11.14 days, 13.18 days, and 14.5 days, respectively. On average for the three periods, 75.15% (2332/3103) of the cancer inpatients had inpatient companions: 62.05% (605/975) in 2019, 83.72% (987/1179) in 2020, and 77.98% (740/949) in 2021; *p* < 0.001. A statistically significant difference was found in the genders of the inpatient companions, (*p* < 0.001), with females (54.59%) outnumbering males (36.96%). A small minority (8.45%) were accompanied by both males and females, and the average number of hours an inpatient companion spent with an inpatient for the three periods was 163.55 ± 279.37 h. Our study conducted further analysis through post hoc tests. The number of hours spent with inpatients in 2019 was higher than those in 2020 and 2021 during the pandemic (Table 1). The four most common departments for cancer inpatients were thoracic (22.08%), hematology-oncology (16.24%), obstetrics (13.47%), and breast surgery (11.57%) (Table 2). There were significant differences in hospital days, nursing hours, and pain scores ≥ 4 between those with and without inpatient companions; *p* < 0.001 (Table 3).

### 3.2. Emotional Assessment Scores of Cancer Inpatients

There were no statistical differences in sleep disturbance, hostility, depression, or inferiority among the cancer inpatients, but there was a statistical difference in anxiety; *p* = 0.040 (Table 1). Cancer inpatients’ emotional scale scores (0–4 points) included sleep disturbance (38.19%), anxiety (30.68%), hostility (21.08%), depression (25.04%), and inferiority (11.15%) (Table 1). In all, 771 cancer inpatients had no inpatient companions and 2332 patients had them. There were significant differences in the emotional scale total scores for those with and without inpatient companions, 1.19 ± 2.18 and 1.93 ± 2.86, respectively; *p* < 0.001. Hostility, depression, and inferiority all showed significant differences, (*p* < 0.001), but no statistical difference was found for gender between those with and without inpatient companions.

## 4. Discussion

Currently, for infection control of the emerging infectious disease of COVID-19, all hospitalized patients are required to register their real names and to be tracked and managed. Therefore, we used this tracking and management data to understand the relationship between cancer inpatients and their inpatient companions. Therefore, the aim of this study was to explore the correlations among the demographic characteristics, pain scores, and emotional stress levels of cancer inpatients and inpatient companions during the COVID-19 pandemic. According to this study, the total emotional scale scores in the anxiety domain in 2019, 2020, and 2021 were 21.13%, 22.14%, and 26.45%, respectively. These figures were significantly different. It can be understood that, as the epidemic situation became more severe, the anxiety levels of the hospitalized patients increased. In addition to worrying about the effects of cancer treatment, they also worried about hospitalization and the risk of nosocomial infection. Further analysis of the relationship between patients with companions and those without companions revealed a significant difference in the total scores for the emotional scales of patients with and without hospital companions. The emotional stress of patients with companions was greater than that of patients without companions. The possible reason is that cancer inpatients have high disease complexity, a greater number of hospitalization days, and fear of the COVID-19 infection. During the impact of the epidemic, psychological stress is more obvious. This finding is similar to those of previous reports [20,21] which identified the fear of contracting COVID-19 and the emotional burden during diagnosis and treatment during the COVID-19 pandemic. As found by Ciążyńska et al. [22], cancer patients often felt stressed due uncertainty regarding their cancer therapy and the risk of developing COVID-19 symptoms while receiving treatment. The COVID-19 pandemic has had an inevitable psychological impact on cancer patients [23]; therefore, there is a need for continued follow-up and psycho-oncological support during this pandemic. In addition, more nursing hours were spent on patients with inpatient companions than on those without inpatient companions. A possible reason is that patients who need inpatient companions have more complex disease conditions. In addition to routine care, nurses also need to respond to problems and the skills of daily care for inpatient companions who accompany patients. According to the literature, inpatient companions who accompany patients can fulfill the role of caring for patients and provide psychological support, which is different from a report from Khaleghparast et al. [24]. The pain scores of inpatients with inpatient companions were higher than those of patients without inpatient companions. A possible explanation is that the inpatient companions were by the patients’ sides 24 h a day, so they could observe the patient and quickly report to the medical staff for immediate treatment.

In our study, the number of nursing hours increased during the pandemic, possibly due to the vacancy periods of inpatient companions who had to wait for the PCR test results. Nursing staff assisted throughout the process, thus increasing the number of daily nursing hours, which is consistent with the literature [9,11]. The primary inpatient companions were 862 males (36.96%) and 1273 females (54.59%). The results of this analysis are consistent with the survey results of the Republic of China Family Caregivers Association, which found that most inpatient companions are female [7,9,25]. The total scores for patients with inpatient companions were higher than those for patients without inpatient companions both before and during the pandemic period. The role of inpatient companions in providing patient support was not shown, unlike in the literature [3,24], but it is also possible that the conditions of the unaccompanied patients were more stable and the scores of each sub-item of the mood scale were lower. There was no statistical difference between male and female patients with or without inpatient companions, but cancer inpatients had a higher rate of female inpatient companions than male inpatient companions during the three periods [7,9].

## 5. Conclusions

Very few studies have focused on inpatient companions in Taiwan. This pandemic provides an opportunity to review the information for our hospitals. Restricting the number of inpatient companions is a necessity under pandemic conditions. When a family member is sick and hospitalized, the family has to try to balance work and life as much as possible. It is often necessary to rotate inpatient companions. Cancer inpatients in our hospital account for 65% of the total inpatients. Most of the cancer inpatients see doctors by appointment. Because of their low immunity, the family members of the patients are worried that they will be infected by the novel coronavirus before and after medical treatment. Although they were more anxious during the pandemic, the total score of the emotional assessment was less than 6 points, which was in the normal range. In 2022, hospitals have also established video consultations, additional video briefing sessions, and family meetings in wards to reduce the frequency of patients and their families visiting doctors and thus the risk of infection.

## 6. Limitations and Future Studies

This study had some limitations. First of all, we obtained the information on visitation policies only from the websites of hospitals in Taiwan. As such, the findings of this study may not be generalizable to hospitals in other regions. Second, with the evolution of COVID-19, hospitals might have changed their visitation policies. Third, we described the hospital visitation policies of ordinary wards in our hospitals in Taiwan. However, the visitation policies for special wards, such as medical and surgical intensive care units, were not included in this study. Therefore, the differences between the visitation policies for ordinary wards and special wards requires further study. Fourth, the hours spent with patients by companions varied widely, ranging from a minimum of 1 h to a maximum of 5967 h. Therefore, in the future we can explore differences in time spent with patients with specific types of cancer. Finally, the pain scores of ≥ 4 points of patients with inpatient companions were higher than those of patients without inpatient companions. Their inpatient companions could observe them 24 h a day and quickly report issues to medical staff for immediate treatment. However, it is impossible to know whether the pain of cancer inpatients is controlled by increases in hospitalization days, and future research should track changes in the pain index.

## Figures and Tables

**Table 1 ijerph-19-07004-t001:** Basic statistics of cancer inpatients (N = 3103) and inpatient companions (N = 2332).

	Total (N = 3103)		Year						*p* Value	Dunn–Bonferroni Post Hoc		
			2019 (*n* = 975)		2020 (*n* = 1179)		2021 (*n* = 949)			2019 vs. 2020	2019 vs. 2021	2020 vs. 2021
Age	59.71	±12.66	59.91	±12.70	60.56	±12.82	58.46	±12.32	<0.001 **	0.571	0.037 *	<0.001 **
Gender									0.058			
Female	1644	52.98%	486	49.85%	645	54.71%	513	54.06%				
Male	1459	47.02%	489	50.15%	534	45.29%	436	45.94%				
Number of days in hospital	7.87	±10.77	8.29	±10.45	7.69	±12.08	7.67	±9.24	<0.001 **	<0.001 **	0.059	0.397
Total nursing hours	12.94	±10.76	11.14	±9.55	13.18	±10.16	14.5	±12.30	<0.001 **	<0.001 **	<0.001 **	0.795
Pain score ≥ 4 points/time	1.45	±6.36	1.53	±7.55	1.42	±6.00	1.41	±5.39	0.113			
Total emotion scale score	1.75	±2.73	1.83	±2.97	1.73	±2.76	1.7	±2.42	0.353			
Sleep disturbance									0.445			
0	1918	61.81%	607	62.26%	740	62.77%	571	60.17%				
1	752	24.23%	233	23.90%	266	22.56%	253	26.66%				
2	308	9.93%	88	9.03%	124	10.52%	96	10.12%				
3	80	2.58%	29	2.97%	31	2.63%	20	2.11%				
4	45	1.45%	18	1.85%	18	1.53%	9	0.95%				
Anxiety									0.040 *			
0	2151	69.32%	692	70.97%	831	70.48%	628	66.17%				
1	718	23.14%	206	21.13%	261	22.14%	251	26.45%				
2	184	5.93%	56	5.74%	67	5.68%	61	6.43%				
3	36	1.16%	15	1.54%	15	1.27%	6	0.63%				
4	14	0.45%	6	0.62%	5	0.42%	3	0.32%				
Hostility									0.849			
0	2449	78.92%	692	70.97%	831	70.48%	628	66.17%				
1	480	15.47%	143	14.67%	174	14.76%	163	17.18%				
2	128	4.13%	37	3.79%	55	4.66%	36	3.79%				
3	31	1.00%	18	1.85%	10	0.85%	3	0.32%				
4	15	0.48%	5	0.51%	6	0.51%	4	0.42%				
Depression									0.216			
0	2326	74.96%	712	73.03%	899	76.25%	715	75.34%				
1	592	19.08%	197	20.21%	204	17.30%	191	20.13%				
2	131	4.22%	40	4.10%	55	4.66%	36	3.79%				
3	38	1.22%	19	1.95%	15	1.27%	4	0.42%				
4	16	0.52%	7	0.72%	6	0.51%	3	0.32%				
Inferiority									0.768			
0	2757	88.85%	863	88.51%	1045	88.63%	849	89.46%				
1	260	8.38%	73	7.49%	103	8.74%	84	8.85%				
2	56	1.80%	22	2.26%	19	1.61%	15	1.58%				
3	21	0.68%	12	1.23%	9	0.76%	0	0.00%				
4	9	0.29%	5	0.51%	3	0.25%	1	0.11%				
Inpatient’s companion’s gender									<0.001 **			
Male	862	36.96%	268	44.30%	320	32.42%	274	37.03%				
Female	1273	54.59%	336	55.54%	532	53.90%	405	54.73%				
Both males and females	197	8.45%	1	0.17%	135	13.68%	61	−8.24%				
Number of hours spent with inpatient	163.55	±279.37	203.37	±359.11	152.87	±281.37	151.28	±207.22	<0.001 **	<0.001 **	<0.001 **	1.000

Kruskal–Wallis test. Chi-Squared test. * *p* < 0.05, ** *p* < 0.01. Continuous data are expressed as mean ± SD.

**Table 2 ijerph-19-07004-t002:** Distribution of cancer inpatients by department.

	Total N = 3103 (%)		Year					
			2019 *n* = 975 (%)		2020 *n* = 1179 (%)		2021 *n* = 949 (%)	
Department								
Chest Medicine	685	(22.08)	216	(22.15)	275	(23.32)	194	(20.44)
Hematology and Oncology	504	(16.24)	156	(16.00)	171	(14.50)	177	(18.65)
Gynecology	418	(13.47)	103	(10.56)	167	(14.16)	148	(15.60)
Breast Surgery	359	(11.57)	115	(11.79)	126	(10.69)	118	(12.43)
Genito-Urinary	235	(7.57)	92	(9.44)	112	(9.50)	31	(3.27)
General Surgery	202	(6.51)	84	(41.58)	73	(36.14)	45	(22.28)
Colorectal Surgery	175	(5.46)	48	(27.43)	60	(34.29)	67	(38.28)
Chest Surgery	132	(4.25)	59	(44.70)	44	(33.33)	29	(21.97)
Gastroenterology	106	(3.43)	25	(23.58)	46	(43.40)	35	(33.02)
Ear, Nose, and Throat	103	(3.32)	38	(36.89)	34	(33.00)	31	(30.11)
Neurosurgery	47	(1.51)	3	(6.38)	20	(42.55)	24	(51.07)
Dentistry	45	(1.45)	20	(44.44)	11	(24.44)	14	(31.12)
Thyroid	40	(1.29)	8	(16.00)	21	(52.50)	11	(31.50)
Others	52	(1.67)	8	(15.38)	19	(36.54)	25	(48.08)

**Table 3 ijerph-19-07004-t003:** Differences between cancer inpatients with and without inpatient companions.

	Total (N = 3103)					2019 (*n* = 975)					2020 (*n* = 1179)					2021 (*n* = 949)				
	Without Inpatient Companions *n* = 771 (%)		With Inpatient Companions *n* = 2332 (%)		*p* Value	Without Inpatient Companions *n* = 370 (%)		With Inpatient Companions *n* = 605 (%)		*p* Value	Without Inpatient Companions *n* = 192 (%)		With Inpatient Companions *n* = 987 (%)		*p* Value	Without Inpatient Companions *n* = 209 (%)		With Inpatient Companions *n* = 740 (%)		*p* Value
Total emotion scale score	1.19	±2.18	1.93	±2.86	<0.001 **	0.99	±1.73	2.34	±3.42	<0.001 **	1.50	±2.97	1.78	±2.71	0.004 **	1.28	±2.01	1.81	±2.51	0.008 **
sleep disturbance	241	31.26	944	(40.4)	<0.001 **	102	(27.5)	266	(43.97)	<0.001 **	63	(32.8)	376	(38.1)	0.192	76	(36.3)	302	(40.8)	0.280
anxiety	170	(22.05)	782	(33.53)	<0.001 **	73	(19.73)	210	(34.71)	<0.001 **	43	(22.4)	305	(30.9)	0.023 *	54	(25.8)	267	(36.0)	0.007 **
hostility	109	(14.14)	545	(23.37)	<0.001 **	39	(10.54)	164	(27.11)	<0.001 **	33	(17.1)	212	(21.4)	0.214	37	(17.7)	169	(22.8)	0.135
depression	134	(17.38)	643	(27.57)	<0.001 **	67	(18.11)	196	(32.40)	<0.001 **	36	(18.7)	244	(24.7)	0.092	31	(14.8)	203	(27.4)	<0.001 **
inferiority	61	(7.91)	285	(12.22)	0.001 **	22	(5.95)	90	(14.8%)	<0.001 **	24	(12.5)	110	(11.1)	0.677	15	(7.18)	85	(11.4)	0.096
Age	56.83	±11.90	60.66	±12.76	<0.001 **	59.67	±12.76	60.05	±12.67	0.534	54.89	±10.18	61.66	±13.00	<0.001 **	53.59	±10.58	59.84	±12.43	<0.001 **
Gender					1.000					0.004 **					0.583					<0.001 **
female	408	(52.92)	1236	(53.00)		162	(43.78)	324	(53.55)		109	(56.7)	536	(54.3)		137	(65.5)	376	(50.8)	
male	363	(47.08)	1096	(47.0)		208	(56.22)	281	(46.45)		83	(43.2)	451	(45.6)		72	(34.4)	364	(49.1)	
Length of stay	4.18	±4.68	9.09	±11.88	<0.001 **	5.23	±5.79	10.16	±12.10	<0.001 **	2.98	±2.21	8.60	±12.97	<0.001 **	3.42	±3.65	8.87	±9.97	<0.001 **
Total nursing hours	8.71	±6.97	14.34	±11.41	<0.001 **	9.34	±7.42	12.23	±10.49	<0.001 **	8.31	±6.11	14.13	±10.51	<0.001 **	7.97	±6.80	16.34	±12.87	<0.001 **
Pain score ≥ 4 points/times	0.51	±2.81	1.76	±7.13	<0.001 **	0.64	±1.87	2.07	±9.44	0.001 **	0.23	±1.29	1.66	±6.51	<0.001 **	0.54	±4.62	1.66	±5.56	<0.001 **

Mann–Whitney U test. Chi-Square test. * *p* < 0.05, ** *p* < 0.01; Continuous data were expressed Mean ± SD.; Categorical data were expressed number and percentage.

## Data Availability

The data presented in this study are available on request from the corresponding author.

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
