# Peer review of "Correlational Study of Emotional Stress, Pain, and the Presence of Inpatient Companions for Cancer Inpatients during the COVID-19 Pandemic"

_ijerph, 2022, doi:10.3390/ijerph19127004_

Round 1

Reviewer 1 Report

Dear Authors,

I have found your paper has clinical important relevance . It is within the scope of the Journal. It suggests that inpatient companion's contribution to cancer patients is immense in parameters.

The methodology is well approached with the appropriate design study. 

I would like to know more about  instruments for measurements like Cronbach α reliability, license, if available.

The use of English is excellent. The literature is current and extensive. Well done!

Author Response

Comment 1. I have found your paper has clinical important relevance. It is within the scope of the Journal. It suggests that inpatient companion's contribution to cancer patients is immense in parameters.
The methodology is well approached with the appropriate design study.
The use of English is excellent. The literature is current and extensive. Well done!

Answer: Thank you for your comments and affirmation.

Comment 2. I would like to know more about instruments for measurements like Cronbach α reliability, license, if available.

Answer: Thank you for your comments
We have added “The BSRS-5 has demonstrated good reliability and validity [18-19]. The internal consistency (Cronbach’s alpha) coefficient of the BSRS-5 was 0.84 [19]” in the revised manuscript. (Lines 109–111)

Original References
18. Lee MB, Liao SC, Lee YJ, Wu CH, Tseng MC, Gau SF, Rau CL: Development and verification of validity and reliability of a short screening instrument to identify psychiatric morbidity. J Formos Med Assoc. 2003, 102: 687-694.
19.Chen HC, Wu CH, Lee YJ, Liao SC, Lee MB: Validity of the five-item brief   symptom rating scale among subjects admitted for general health screening. J Formos Med Assoc. 2005, 104: 824-829.

Reviewer 2 Report

Overall review

This is an interesting study investigating the emotional stress and pain of cancer inpatients following the presence of inpatient companions during the covid-19 pandemic.

Minor revision

Statistical analysis

The Kruskal-Wallis rank sum test was used as a statistical analysis method, but there is no description of the post-test results.

Results

Line 163-165

Change table 4 to table 1Write emotional scale score (1-4 points) in the column of table1.

Line 176-177

It is 176 likely that the support and company of family members made the patient more likely to 177 show dependence and depressed mood

- Additional explanation is needed for the above sentence

Line 179-180

In addition, more nursing hours were spent on patients with inpatient companions 179 than on those without inpatient companions.

Information on nursing hours was not found in Table 3.

Limitations and Future Studies

Line 232-236

A description of the need for future research in nursing care is considered to be a statement unrelated to the theme of this study

Author Response

Comment 1. This is an interesting study investigating the emotional stress and pain of cancer inpatients following the presence of inpatient companions during the covid-19 pandemic.

Answer: Thank you for the encouragement. 

Comment 2.
Statistical analysis
The Kruskal-Wallis rank sum test was used as a statistical analysis method, but there is no description of the post-test results. Line 163-165

Answer:
Regarding the post-test results section, we have re-run the statistical analysis and re-written the results and discussion. Lines 122–124, 133–144, and 152–156.

Comment 3. Change table 4 to table 1Write emotional scale score (1-4 points) in the column of table1. Line 176-177

Answer: Thank you for your comments.
We have changed Table 4 to Table 1. Line 161.

Comment 4.
It is 176 likely that the support and company of family members made the patient more likely to 177 show dependence and depressed mood.
Additional explanation is needed for the above sentence. Line 179-180

Answer:
Thank you for your comments.
We have revised this in the MS. Lines 207-220.

Comment 5. In addition, more nursing hours were spent on patients with inpatient companions than on those without inpatient companions.

Information on nursing hours was not found in Table 3.

Answer:
Thank you for your comments.
We have changed patient to total nursing hours in Table 3. (Line 23)

Comment 6.
 Limitations and Future Studies Line 232-236.
A description of the need for future research in nursing care is considered to be a statement unrelated to the theme of this study

Answer: Thank you for your comments. We have deleted this sentence from Lines 279–283.
